# Mutation-Driven S100A8 Overexpression Confers Aberrant Phenotypes in Type 1 *CALR*-Mutated MPN

**DOI:** 10.3390/ijms24108747

**Published:** 2023-05-14

**Authors:** Ying-Hsuan Wang, Ying-Ju Chen, Yi-Hua Lai, Ming-Chung Wang, Yi-Yang Chen, Yu-Ying Wu, Yao-Ren Yang, Hsing-Yi Tsou, Chian-Pei Li, Chia-Chen Hsu, Cih-En Huang, Chih-Cheng Chen

**Affiliations:** 1Division of Hematology and Oncology, Department of Medicine, Chang Gung Memorial Hospital, Chiayi 61363, Taiwan; shang0320@cgmh.org.tw (Y.-H.W.); agrb7289@cgu.edu.tw (Y.-J.C.); hbm486426@gmail.com (Y.-H.L.); 8902025@cgmh.org.tw (Y.-Y.C.); yuyun5801@cgmh.org.tw (Y.-Y.W.); b9802046@cgmh.org.tw (Y.-R.Y.); a6986a@gmail.com (H.-Y.T.); air110202@cgmh.org.tw (C.-P.L.); loofah1008@cgmh.org.tw (C.-C.H.); 2Division of Hematology and Oncology, Department of Medicine, Chang Gung Memorial Hospital, Kaohsiung 83301, Taiwan; wangmt@cgmh.org.tw; 3College of Medicine, Chang Gung University, Taoyuan 33302, Taiwan

**Keywords:** myeloproliferative neoplasms, calreticulin, type 1 *CALR* mutation, S100A8, promoter hypomethylation, phenotype

## Abstract

Numerous pathogenic *CALR* exon 9 mutations have been identified in myeloproliferative neoplasms (MPN), with type 1 (52bp deletion; *CALR*^DEL^) and type 2 (5bp insertion; *CALR*^INS^) being the most prevalent. Despite the universal pathobiology of MPN driven by various *CALR* mutants, it is unclear why different *CALR* mutations result in diverse clinical phenotypes. Through RNA sequencing followed by validation at the protein and mRNA levels, we found that S100A8 was specifically enriched in *CALR*^DEL^ but not in *CALR*^INS^ MPN-model cells. The expression of *S100a8* could be regulated by STAT3 based on luciferase reporter assay complemented with inhibitor treatment. Pyrosequencing demonstrated relative hypomethylation in two CpG sites within the potential pSTAT3-targeting *S100a8* promoter region in *CALR*^DEL^ cells as compared to *CALR*^INS^ cells, suggesting that distinct epigenetic alteration could factor into the divergent S100A8 levels in these cells. The functional analysis confirmed that S100A8 non-redundantly contributed to accelerated cellular proliferation and reduced apoptosis in *CALR*^DEL^ cells. Clinical validation showed significantly enhanced *S100A8* expression in *CALR*^DEL^-mutated MPN patients compared to *CALR*^INS^-mutated cases, and thrombocytosis was less prominent in those with *S100A8* upregulation. This study provides indispensable insights into how different *CALR* mutations discrepantly drive the expression of specific genes that contributes to unique phenotypes in MPN.

## 1. Introduction

Classical Philadelphia chromosome-negative (Ph-neg) myeloproliferative neoplasms (MPN) are clonal hematopoietic disorders with a unique pathogenic mechanism in which mutually exclusive driver mutations in *calreticulin* (*CALR*), *MPL*, and *JAK2* trigger the activation of Janus Kinase/Signal Transducer and Activator of Transcription (JAK/STAT) signaling axis [1]. Polycythemia vera (PV), essential thrombocythemia (ET), and primary myelofibrosis (PMF) (including prefibrotic and overt PMF) constitute the most common subtypes of Ph-neg MPN [2]. However, it is perplexing that different disease subtypes may develop among patients with an identical driver mutation, and patients with the same disease subtype may carry disparate driver mutations or even run distinct clinical courses regarding disease-related complications and rates of transformation. Ongoing efforts have been made to dissect the enigmas behind the molecular pathogenesis that leads to phenotypic specification in MPN.

Pathogenic *CALR* mutations in MPN consist of +1 base-pair frameshift insertions and/or deletions in exon 9, leading to the generation of a novel mutant-specific, positively charged amino acid sequence with accompanying loss of the C-terminal endoplasmic reticulum (ER)-retention signal KDEL motif [3]. Mutant CALR proteins possess an increased affinity to the thrombopoietin receptor MPL and interaction between them results in a constitutive JAK/STAT signaling activation [3]. Numerous subtypes of *CALR* mutations have been identified in MPN ever since their discovery, with type 1 (52bp deletion; L367fs*46; *CALR*^DEL^) and type 2 (5bp “TTGTC” insertion; K385fs*47; *CALR*^INS^) mutations being the most prevalent [4,5]. *CALR*-mutated MPN patients do exhibit dissimilar clinical characteristics from those with either *JAK2* or *MPL* mutations [3]. However, in spite of the fact that both types of *CALR* mutations share some molecular similarities between those MPN cells, patients with different *CALR* mutations still have disparate clinical trajectories [6,7]. For example, type 2 *CALR*-mutated patients are more likely to have higher platelet counts (in those with ET subtype) [7] and own a more adverse prognosis while clustering with more frequent epigenetic mutations (in those with MF subtype) [6]. It is considered that the secondary structure of various mutant CALR proteins underlies the heterogeneity in their respective phenotypes. To recapitulate structural resemblance among some variants, those CALR mutants are further subcategorized into “type 1-like” and “type 2-like” based on the length of negatively charged amino acid stretches deleted in these mutants [8,9]. Studies focusing on the mechanism of *CALR*-mutated MPN have shown that these two types of CALR mutants display differential loss of function, as type 1 CALR loses its Ca^2+^-binding activity which leads to activation of IRE1α-XBP1 pathway of the unfolded protein response (UPR) [10], whereas type 2 CALR loses its molecular chaperone ability, which results in activation of independency on ATF6 pathway of the UPR [11]. In spite of that, information on transcriptomic discrepancies leading to distinct molecular mechanisms with phenotypic implications between type 1 and type 2 CALR mutations is lacking.

S100A8 and S100A9 are calcium-binding proteins that belong to the S100 family of myeloid-related proteins [12]. The S100A8/A9 heterocomplex has been known as a ligand for both toll-like receptor 4 (TLR4) and receptor for advanced glycation end products (RAGE) to exert its effect on activation of downstream extracellular-regulated kinase 1/2 (ERK1/2) and phosphatidylinositol 3-kinase (PI3K)/AKT signaling pathways [13]. They play significant roles in a variety of biological processes, including regulation of cell growth and differentiation, immune response, and inflammation [14]. Elevated levels of S100A8/A9 have been implicated in several diseases, including cancers and autoimmune disorders [15].

There has been a long history between MPN and systemic inflammation. Chronic inflammation per se has been considered a trigger for the development and subsequent clonal evolution of myeloid neoplasms, including MPN [16]. MPN, as a disease, creates an accommodating niche in the bone marrow through the induction of systemic inflammation to facilitate its propagation [17]. S100A8/A9, capable of supporting the myeloproliferation during chronic inflammation [18], has been linked to MPN as well. A study exploring the role of *EZH2* mutation in MPN pathogenesis inadvertently identified up-regulation of S100A8 and S100A9 in *EZH2*-deleted, *JAK2*-mutated MPN-model cells [19]. Through single-cell RNA sequencing (scRNA-Seq) coupling with *JAK2*-mutated murine models, investigators have demonstrated that pre-fibrotic mesenchymal stem cells express the S100A8/A9 complex to promote MF development through pro-inflammatory signaling [20]. Intriguingly, there is no established correlation between S100A8/A9 and *CALR*-mutated MPNs. 

In our previous work, we employed scRNA-Seq to explore the transcriptomic profiling in clinical MPN samples and identified *S100A8* as one of several differentially expressed genes between *JAK2*- and *CALR*-mutated patients [21]. We speculated that S100A8/A9 might play some role in the pathogenesis of *CALR*-mutated MPN. Here, we reveal that S100A8 is particularly enriched in type 1 *CALR*-mutated MPN-model cells. Functional analyses further clarify the regulatory mechanism of S100A8 overexpression and confirm the non-redundancy of its role in the cellular characteristics of *CALR*^DEL^-mutated MPN. Our data provide novel insights into the phenotypic heterogeneity of MPN driven by discrepant *CALR* mutations.

## 2. Results

### 2.1. Establishment of CALR-Mutated MPN-Model Cells

To dissect the transcriptomic discrepancies between type 1 and type 2 *CALR*-mutated MPN, we aimed to use the IL-3-dependent Ba/F3 (BaF3) cell line to establish MPN cell models, which ectopically expressed respective mutant CALRs. With the indispensable role of MPL as an interacting partner for the mutated CALR proteins to exert their oncogenic potential [22,23] and with the lack of *MPL* expression in BaF3 cells, we had to overexpress wild-type *MPL* (*MPL*^WT^) in those *CALR*-mutated (*CALR*^MUT^) MPN-model cells as well. These cell models were established through the co-transduction of an *MPL*^WT^-expressing lentiviral vector and an MSCV vector cloned with either *CALR*^WT^, *CALR*^INS^, or *CALR*^DEL^ into BaF3 cells (Figure 1A). Desirable stable clones were purified with antibiotics selection and verified with flow cytometric analysis (Figure 1B). Western blotting (WB) confirmed the presence of target proteins (Figure 1C). Importantly, activation of the JAK/STAT signal pathway was exclusively seen in both *CALR*-mutated clones (Figure 1D). Furthermore, these *CALR*-mutated cells exhibited accelerated, IL-3-independent growth and significantly less apoptosis in the absence of IL-3 as compared to the parental control (Figure 1E,F). The results demonstrated the adequacy of our established cell lines as in vitro MPN models for our subsequent experiments in this work.

### 2.2. CALR^DEL^ Cells but Not CALR^INS^ Cells Exhibit Unequivocal Up-Regulation of S100a8

To explore the potential transcriptomic discrepancy between type 1 and type 2 *CALR*-mutated MPN-model cells, we performed RNA sequencing (RNA-Seq) in *CALR*^WT^, *CALR*^INS^, and *CALR*^DEL^ BaF3 cells. As demonstrated in Figure 2A, both *S100a8* and *S100a9* were found to be among the most significantly enriched genes in *CALR*^DEL^ cells as compared to wild-type cells, an observation not identified in *CALR*^INS^ cells. Quantitative RT-PCR (qRT-PCR) analyses further confirmed the differential expression pattern of *S100a8/a9* between *CALR*^DEL^ and *CALR*^INS^ cells (Figure 2B), with *S100a8* overexpression being more pronounced than *S100a9* in *CALR*^DEL^ cells.

Intrigued by these findings, we reasoned that distinct *CALR* mutations might have led to differential activation of S100A8/A9 and their downstream signaling pathways. Therefore, we assessed the protein levels of S100A8/A9, RAGE, TLR4, and phosphorylated AKT with WB. Although there was a slight difference with regards to the expression of S100A8 (also up-regulated in *CALR*^INS^ cells) at the protein level, the enrichment of S100A8 protein in *CALR*^DEL^ cells was in line with the results from RNA-Seq and qRT-PCR analyses (Figure 2C). On the other hand, there were no drastic changes in TLR4 and RAGE expression in MPN-model cells, suggesting that these downstream signalings were not specifically activated. Regarding the protein expression of S100A9, we were unable to obtain a clear band similar to S100A8. This could be due to the low expression levels of S100A9 in these cells, making it hard to detect on WB. Given that both S100A8 and S100A9 function as secretory proteins, we also quantified their levels in the cell culture supernatants. Through enzyme-linked immunosorbent assay (ELISA, Helsinki, Finland), we found that *CALR*^DEL^ cells (but not *CALR*^INS^ cells) exhibited enhanced secretion of both S100A8 and S100A9 (Figure 2D). Noteworthily, the increment of secreted S100A8 in *CALR*^DEL^ cells was more remarkable (about three times higher than that of the wild-type cells), whereas there was only a slight elevation of secreted S100A9 in these cells.

Taken together, these results confirm that *CALR*^DEL^ cells, but not *CALR*^INS^ cells, specifically exhibit up-regulation of S100A8. Whether such an enrichment plays a role in determining the phenotypic heterogeneity between type 1 and 2 *CALR*-mutated MPN warrants further investigation.

### 2.3. Inhibition of JAK/STAT3 Signaling Suppresses S100A8 Expression in CALR^DEL^ MPN

With constitutive JAK/STAT activation being a hallmark phenomenon in MPN, we wondered if the signaling could be involved in the up-regulation of S100A8 expression in *CALR*^DEL^ MPN-model cells. Upon treatment with a STAT3 inhibitor S3I-201 at a concentration of 25 μM, we found that S100A8 protein was significantly reduced in these cells (Figure 3A). There was a drastic decrease in the secreted S100A8 level in the cell culture supernatant as well (Figure 3B). A cytotoxic assay justified the adequacy of the experimental concentration we used, as it appropriately suppressed phosphorylated STAT3 in those *CALR*^DEL^ cells (Figure 3A) while sparing their viability (Figure 3C), indicating the repression of S100A8 was not a result of the cytotoxicity of the STAT3 inhibitor. These data suggest that JAK/STAT3 signaling activity non-redundantly mediates the expression of S100A8 in *CALR*^DEL^ cells. It also implies that JAK/STAT3 pathway could be equivalently involved in the S100A8-mediated pathogenesis of MPN provoked by *CALR*^DEL^ mutation.

### 2.4. STAT3 Regulates S100a8 Expression in a Cell Type-Specific Manner, with Disparate Promoter Methylation Being One Potential Mechanism Involved in the Heterogeneous Regulation between CALR^DEL^ and CALR^INS^ Cells

To further investigate if STAT3 regulated the transcription of *S100a8*, we employed luciferase reporter assay. We cloned the *S100a8* promoter sequence into a luciferase reporter vector (Figure 4A) to evaluate whether STAT3 exerted its effect through interaction with the *S100a8* promoter. Upon transfection, the luciferase activity induced by the cloned *S100a8* promoter sequence was significantly higher in *CALR*^DEL^ cells than that seen in the empty vector (EV) control cells, whereas no meaningful disparity could be observed in both *CALR*^WT^ and *CALR*^INS^ cells (Figure 4B). Moreover, concurrent treatment with STAT3 inhibitor S3I-201 suppressed the enhanced luciferase activity in *CALR*^DEL^ cells (Figure 4C). With all driver mutations (including both type 1 and type 2 *CALR* mutations) of MPN led to constitutive activation of JAK/STAT signaling, our results demonstrating that STAT3 preferentially regulated the transcription of *S100a8* in *CALR*^DEL^ cells but not in *CALR*^INS^ cells were rather interesting. It is well known that DNA methylation plays a crucial role in regulating gene expression through either recruiting proteins involved in gene repression or inhibiting the binding of transcription factors to the DNA [24]. This raises the possibility that the degree of DNA methylation at the *S100a8* promoter region could be the difference maker in impacting its discrepant expression across MPN-model cells with distinct driver mutations. To substantiate this, bisulfite pyrosequencing was performed to clarify the methylation status of the *S100a8* promoter region. The results showed hypomethylation at two CpG sites (positions 1 and 3) in *CALR*^DEL^ cells as compared to both *CALR*^WT^ and *CALR*^INS^ cells (Figure 4D). For the remaining CpG positions, the degree of methylation was comparable between *CALR*^INS^ and *CALR*^DEL^ cells. The data suggest that disparate regulation and expression of *S100a8* across different *CALR*-mutated MPN-model cells might result from their unique methylation landscape. The *S100a8* gene, being transcriptionally regulated by STAT3 in *CALR*^DEL^ MPN-model cells, possesses more profound promoter hypomethylation within its potential STAT3 binding regions and is exclusively upregulated in these cells.

### 2.5. S100A8 Nonredundantly Contributes to Accelerated Cellular Proliferation and Reduced Apoptosis in CALR^DEL^ Cells

To further determine if S100A8 was involved in *CALR*^DEL^-induced pathogenesis of MPN, we knocked down the expression of *S100a8* in *CALR*^DEL^ cells using five independent short hairpin RNAs (shRNAs) targeting *S100a8*. As determined by WB, all five *S100a8* shRNAs (sh*S100a8*) achieved desirable effects in down-regulating the expression of S100A8 (Figure 5A). Among all, sh*S100a8*-A was chosen for subsequent experiments to evaluate potential phenotypic changes in *CALR*^DEL^ cells. S100A8 expression manifested as either a cellular protein or a secreted cytokine, was unequivocally reduced by the *S100a8*-specific shRNA (Figure 5A,B). Furthermore, *S100a8* knockdown significantly inhibited the viability of these *CALR*^DEL^ cells (Figure 5C). We then employed flow cytometric analyses on various markers to discriminate if the diminished cellular viability was the result of either declined proliferation or excessive apoptosis. Compared to those exposed to scramble sequence-targeting shRNA (shSC), *CALR*^DEL^ cells treated with sh*S100a8*-A expressed drastically decreased proliferation marker Ki67 (Figure 5D) and underwent more apoptosis (Figure 5E). There was no apparent difference with regard to early apoptosis, but those *S100a8*-knockdown *CALR*^DEL^ cells exhibited enhanced late apoptosis (Figure 5E). Upon staining with apoptotic markers, those sh*S100a8*-A-treated cells expressed more cleaved caspase 9, cleaved caspase 3, and cleavage form of poly (ADP-ribose) polymerase (PARP) (Figure 5F). These results indicate that S100A8 plays an important role in mediating cellular proliferation and survival in *CALR*^DEL^ cells.

### 2.6. S100A8-Overexpressing MPN Patients Exhibit Unique Clinical Characteristics

To appraise the relevance of *S100A8* in MPN patients with distinct driver mutations (*JAK2*^V617F^, *CALR*^INS^, *CALR*^DEL^), we sought validation from clinical samples. Previous studies have shown that the length of negatively charged amino acid stretches deleted in various CALR mutants is associated with distinct clinical features. Based on structural resemblance and phenotypic similarities, we followed the Mayo classification guide and subcategorized our *CALR-*mutated patients into two major groups, type 1- and type 2-like variants [9]. Quantitative RT-PCR was employed to measure the expression levels of *S100A8* in the peripheral blood (PB) granulocytes of MPN patients and the healthy control normal population (NP). While type 2-like *CALR*-mutated patients did not exhibit aberrant *S100A8* expression in their PB granulocytes as compared to NP, it was conspicuous that the mean *S100A8* levels in both *JAK2*- and type 1-like *CALR* mutated cases were significantly higher than the control population (Figure 6A). This was consistent with our cell line data demonstrating enriched *S100a8* expression in *CALR*^DEL^ cells (Figure 2B).

Meanwhile, we noticed that not all type 1-like *CALR*-mutated MPN patients possessed escalated *S100A8* levels (Figure 6A), indicating the presence of inter-individual heterogeneity. We, therefore, further divided those patients into *S100A8* high- and low-expressing categories based on whether the expression of this gene exceeded the mean level of the whole control population or not. Table 1 demonstrated the clinical and laboratory characteristics of the 31 type 1-like *CALR*-mutated MPN patients sub-stratified into *S100A8*^High^ and *S100A8*^Low^ groups. Although most parameters did not differ drastically between the two groups of patients, it was interesting to see that *S100A8*^High^ patients had seemingly less severe thrombocytosis than *S100A8*^Low^ patients (625 ± 318 × 10^9^/L vs. 975 ± 527 × 10^9^/L, Table 1). Unfortunately, probably hampered by limited case numbers, such a comparison was not statistically significant (*p* = 0.077). In a subsidiary comparison, our type 1-like *CALR*-mutated patients also had lower platelet counts than their counterparts, which is in line with previous reports indicating that MPN patients with type 2 *CALR* mutation have more prominent thrombocytosis than those with type 1 mutation [7,25]. With *S100A8* overexpression being more profoundly enriched in type 1-like *CALR*-mutated patients and with higher *S100A8* expression being correlated with lower platelet counts in our patients, it would be intriguing to see if *S100A8* could be particularly pertinent in the phenotypic heterogeneity of MPN propagated by different subtypes of *CALR* mutation.

With regards to MPN-associated catastrophic events, *S100A8*^High^ type 1-like *CALR*-mutated MPN patients were not more likely to suffer from major thrombotic events, secondary MF, or leukemia transformation (Table 1). In line with this observation, all the survival outcomes (including thrombosis-free survival, MF-free survival, and overall survival) were comparable between *S100A8*^High^ and *S100A8*^Low^ patients (Figure 6B). This probably reflects the relatively indolent nature of type 1-like *CALR*-mutated MPN. The limited cases enrolled for this study certainly did not help either, as the small number of patients would not have been ample enough to identify any potential differences in these regards.

## 3. Discussion

*CALR* mutations are the most unique among the three main driver mutations in MPN. Unlike *JAK2* and *MPL* mutations, which are found in the genes coding for either a receptor tyrosine kinase or a hematopoietic growth factor receptor, these mutations occur in a gene that encodes an ER chaperon protein that regulates intracellular calcium homeostasis and is seemingly irrelevant to hematopoiesis in its normal function. Additionally, those mutations include numerous subtypes, but all of them lead to a novel, concordant C-terminal peptide sequence with enhanced interacting affinity to MPL, which results in accelerated hematopoietic activity. The sub-stratification of type 1-like and type 2-like mutations successfully classifies *CALR*-mutated MPN patients into two representative entities with phenotypes that are more or less distinctive [26]. However, although some studies have identified contrasting pathogenesis between them [10,26], it remains a major enigma of how and why these two classes of variants drive the development and propagation of diseases disparately. Our work, through RNA-Seq, followed by in vitro and clinical validations, provides an innovative perspective on how S100A8 may be differentially activated in type 1 *CALR*-mutated MPN, a finding that could offer potential implications for therapeutic and prognostic purposes.

The initial data on increased S100A8 expression exclusively seen in *CALR*^DEL^ MPN-model cells, identified on RNA-Seq and validated through western blotting and ELISA analyses, was really intriguing to us. It has been confirmed that JAK/STAT3 signaling mediates the expression of S100A8 [27,28]. Given that the JAK/STAT pathway is equivalently activated in all *CALR*-mutated subtypes, we were surprised to see that the *S100A8* promoter was not induced in *CALR*^INS^ cells in the luciferase reporter assay (Figure 4B). The assay per se could not demonstrate an effect of STAT3-mediated regulation of S100A8 in *CALR*^DEL^ cells, but subsequent experiments on combined treatment with a STAT3 inhibitor indirectly supported such a probability (Figure 4C). Substantially, the lesser degree of methylation within certain CpG sites of the *S100A8* promoter in *CALR*^DEL^ cells suggested that discrepant epigenetic alteration could be the key to the divergent S100A8 levels between the two types of mutated cells. Interestingly, it has been well-documented that increased epigenetic mutations are more commonly seen in type 2 *CALR*-mutated MF patients [6]. In a recent study in patients with myelodysplastic syndrome, a closely related myeloid neoplasm with overlapping genetic alterations with MPN, the investigators observed altered expression of S100A8 in cases with epigenetic mutations [29]. These data are in concordance with our discovery of more profound promoter hypermethylation in *CALR*^INS^ cells. Comparative studies through targeted sequencing and methylation sequencing would further clarify the epigenetic diversity between type 1 and type 2 *CALR*-mutated cells. Unfortunately, reports on this aspect are lacking.

Other than aberrant upregulation, S100A8 also imposed a significant influence on cellular viability and apoptosis in our *CALR*^DEL^ MPN cell model. S100A8 has been implicated in the proliferation, differentiation, and apoptosis of various cell types, suggesting its essential role in cellular growth. While there is ample evidence showing that S100A8 possesses tumorigenic potentials in many solid tumors, it has also been associated with an increased risk of hematological malignancies, especially myeloid neoplasms [12]. Collectively, through actions on immune response, oxidative stress, and cell growth, S100A8 helps shape the microenvironment that favors the clonal expansion of the myeloid neoplasms [12,30,31]. There have been some recent reports demonstrating an association between S100A8 and MPN as well. The S100A8 plasma levels were found to be increased in MPN patients [32], whereas our study specifically revealed that S100A8 was more pronouncedly overexpressed in *CALR*^DEL^-mutated cases. It is well-known that type 1-like *CALR*-mutated ET patients are at a greater risk of developing post-ET MF [26]. Could the increased S100A8 level confer the excessive risk for MF transformation in these patients? The S100A8 proteins are known as alarmins, and their expression is increased in various cells during inflammatory conditions, leading to elevated serum levels of S100A8 in many inflammatory disorders [12]. Alternatively, S100A8 further amplifies inflammatory response by facilitating the secretion of pro-inflammatory cytokines [12]. Considering inflammation is a well-documented provoking factor for MF formation or transformation in MPN [33], it is, therefore, plausible that S100A8 upregulation creates a more inflammatory niche, which, in turn, serves as the driving force for the progression of bone marrow fibrosis. The hypothesis has already been validated in a *JAK2*-mutated murine model [20], in which the expression of S100A8/A9 heterocomplex provoked inflammation and promoted myelofibrosis, both of which could be reversible upon treatment with a small molecule targeting S100A8/A9 signaling inhibition [20]. Whether these findings are applicable to type 1 *CALR*-mutated MPN remains to be elucidated.

Hampered by the limited number of *CALR*-mutated MPN cases in our study cohort, we could not identify any prognostic relevance of *S100A8* overexpression in these patients (Table 1 and Figure 6B). Exceptionally, there was a trend for *S100A8*^Low^ patients to have more profound thrombocytosis than their counterparts (Table 1). With increased S100A8 expression more significantly enriched in our type 1-like *CALR-*mutated patients and with these patients being known to be less thrombocythemic [7], it makes us wonder if S100A8 is also actively involved in altered thrombopoiesis in *CALR*-mutated MPN. Yang et al. previously demonstrated that overexpression of *S100a8* increases megakaryocytic colonies in the bone marrow of *Jak2*V617F mice [19], yet contrasting evidence has shown that S100A8/A9 heterocomplex activates TLR4 in mediating MF development while also causing dysregulated megakaryopoiesis [34]. The relationship between S100A8 signaling axis and thrombopoiesis could be best explored in *CALR*-mutated murine models, which would also help clarify several key questions originated from this work, including whether increased S100A8 does accelerate MF transformation, whether S100A8 plays an essential role in *CALR*^DEL^ mutation-associated MPN pathogenesis while it becomes dispensable in *CALR*^INS^ mutation-induced disease propagation, and whether S100A8 could create a more inflammatory milieu in the *CALR*^DEL^-mutated BM. These questions are, unfortunately, unaddressed in our current study.

In conclusion, we have demonstrated that type 1 but not type 2 *CALR-*mutated MPN exhibits unequivocal S100A8 overexpression. Indirect evidence from our experiments suggests that JAK/STAT signaling (specifically STAT3) could be pertinent in the regulation of S100A8 expression, whereas discrepant epigenetic changes could be the key to the divergent S100A8 levels between type 1 and type 2 *CALR-*mutated MPN cells. S100A8 no-redundantly contributes to accelerated cellular proliferation and reduced apoptosis in *CALR*^DEL^ cells. The discovery of a link between S100A8 and type 1 *CALR*-mutated MPN represents a significant advancement in the understanding of phenotypic heterogeneity in patients with different subtypes of *CALR* mutations.

## 4. Materials and Methods

### 4.1. Cell Lines, Culture, and Viral Transduction

*CALR*^WT^, *CALR*^DEL^, and *CALR*^INS^ cells were established through the co-transduction of a human MPL-expressing lentiviral vector and an MSCV vector (constructed with either wild-type *CALR*, type 1 *CALR*, or type 2 *CALR* mutations) into BaF3 cells. The mutant-carrying plasmids were kindly provided by Professor Tony R. Green from the University of Cambridge, UK. Stable colonies were selected and maintained in a medium containing puromycin (2.5 μg/mL) and G418 (0.5 mg/mL). The genomic DNA of stable cells was extracted and validated by construct-specific primer sets in all cell lines. Western blot and flow cytometry were employed for validation. Parental cells and BaF3 cells transduced with either MPL + EV or MPL + *CALR*^WT^ were maintained in culture media containing 2 ng/mL IL-3 until 16 h before experiments when IL-3 was removed. Meanwhile, *CALR*^INS^ and *CALR*^DEL^ cells were maintained in IL-3-free media. All cells were maintained in RPMI-1640 medium (CORNING) supplemented with 10% fetal bovine serum (ThermoFisher) and Antibiotic Antimycotic Solution (100×) (SI-A5955, Sigma) at 37 °C in a humidified atmosphere containing 5% CO_2_.

To knock down *S100a8* expression, lentiviruses expressing distinct short hairpin RNA (shRNA) were purchased from the National RNAi Core Facility, Academia Sinica, Taiwan. Five different *S100a8* shRNA target sequences were designed as follows: sh*S100a8*-A, 5′-TGCAATTAACTTCGAGGAGTT-3′; sh*S100a8*-B, 5′-CACTACTGAGTGTCCTCAGTT-3′; sh*S100a8*-C, 5′-TCCTCAGTTTGTGCAGAATAT-3′; sh*S100a8*-D, 5′-TCAGAGAATTGGACATCAATA-3′; and sh*S100a8*-E, 5′-CAACCTCATTGATGTCTACCA-3′.

### 4.2. RNA-Seq, qRT-PCR, and ELISA

RNA sequencing was performed as previously described [35]. In brief, following RNA extraction (from *CALR*^WT^, *CALR*^DEL^, and *CALR*^INS^ cells), library construction, and sequencing on an Illumina sequencing platform, the FASTQ data files, quality control examination, and sequencing trimming processes were generated through Welgene’s pipeline based on Illumina’s base-calling program bcl2fastq v2.2.0 (WELGENE Biotech, Taipei, Taiwan). Analysis of differentially expressed genes with genome bias detection/correction was performed on Cuffdiff [36] and Welgene in-house programs (WELGENE Biotech, Taipei, Taiwan).

For qRT-PCR analysis, total RNA was isolated from indicated cell lines or patients’ peripheral blood granulocytes using the TOOLSmart RNA Extractor (TOOLS). First-strand complementary DNA (cDNA) was generated from 1 μg of RNA using a TOOLSQuant II Fast RT Kit (TOOLS). Gene expression levels were measured via qRT-PCR with a Rotor-Gene Q real-time PCR detection system (QIAGEN) using GoTaq^®^ qPCR Master Mix (Promega, Madison, WI, USA), and the results were normalized to GAPDH. The relative expression level of each target gene was calculated using the 2^−ΔΔCT^ method as previously described [37]. All qRT-PCR amplifications were performed in triplicate, and the experiments were repeated at least three times.

For quantification of secreted protein levels, the secreted S100A8 and S100A9 were measured with ELISA kits (tcue2854 and tcue2855, Taiclone) according to the manufacturer’s instructions.

### 4.3. Cell Viability, Proliferation, and Apoptosis Assays

Cell viability was measured with XTT assay according to the manufacturer’s protocol (Biological Industries, Beit Haemek, Israel). For the STAT3 inhibition assay, STAT3 Inhibitor S3I-201 (573102, Sigma-Aldrich, Macquarie, Australia) was dissolved in dimethyl sulfoxide (DMSO) at a concentration of 25 mM as the stock solution. *CALR*^DEL^ cells were exposed to various concentrations of S3I-201 for 48 h, and the IC_50_ concentration was determined by XTT assay. Based on the obtained information, cells were treated with S3I-201 at a concentration of 25 µM that did not affect the cellular viability in subsequent STAT3 inhibition assays.

To assess cellular proliferation, cells cultured for 24 h were harvested for Ki67-staining (BioLegend, San Diego, CA, USA) and subsequent flow cytometric analysis. The percentages of apoptotic cells were determined using a FACSCanto II flow cytometer (BD Biosciences, San Jose, CA, USA) after the cells were treated with a 7-AAD and APC-Annexin V Apoptosis Detection Kit (BD Biosciences). All experiments were performed in triplicate, and only representative data were shown.

### 4.4. Western Blot

Cells were homogenized in RIPA lysis buffer (Thermo Scientific, Waltham, MA, USA). Protein was separated by SDS-PAGE, immunoblotted with the indicated antibodies, and visualized using the ECL. Antibodies used were as follows: anti-CALR (208912, Sigma-Aldrich, St Louis, MO, USA), anti-CALR^Mut^ (DIA-CAL-250, Dianova, Kanjirappally, Kerala, India), anti-MPL (06-944, Sigma-Aldrich, St Louis, MO, USA), anti-FLAG (F3165, Sigma-Aldrich, St Louis, MO, USA), anti-β-Actin (20536-1-AP, Proteintech, Illinois, USA), anti-JAK2 (04-001-25UG, Millipore, Burlington, MA, USA)), anti-p-JAK2 (3771, Cell Signaling, Danvers, MA, USA), anti-STAT3 (12640, Cell Signaling, Danvers, MA, USA), anti-p-STAT3 (9145, Cell Signaling, Danvers, MA, USA), anti-S100A8 (tcea21402, Taiclone, Taipei, Taiwan), anti-S100A9 (tcba12809, Taiclone, Taipei, Taiwan), anti-RAGE (tcea21621, Taiclone, Taipei, Taiwan), anti-TLR4 (ARG54702, Arigo Biolaboratories, Hsinchu, Taiwan), anti-AKT (4691, Cell Signaling, Danvers, MA, USA), anti-p-AKT (4060, Cell Signaling, Danvers, MA, USA), anti-Caspase 9 (SAB3500405, Sigma-Aldrich, St Louis, MO, USA), anti-Caspase 3 (C8487, Sigma-Aldrich, St Louis, MO, USA), and anti-PARP (9542, Cell Signaling, Danvers, MA, USA).

### 4.5. Luciferase Reporter Assay

The full-length promoter region of *S100a8* (−941/+76 bp upstream from the transcriptional start site) was amplified and cloned into the pGL4.26 Firefly luciferase reporter construct. The Firefly constructs with or without *S100a8* promoter were co-transfected with pGL4.74 Renilla luciferase constructs into 1 × 10^6^ *CALR*^WT^, *CALR*^DEL^, and *CALR*^INS^ cells using Amaxa^®^ Cell Line Nucleofector^®^ Kit V (Lonza), respectively. After 48 h, the luciferase activity was measured with the Dual-Glo^®^ Luciferase Assay System (Promega, Madison, WI, USA). For each sample, the ratio of Firefly to Renilla luciferase activity was quantified and normalized to the empty vector.

### 4.6. Bisulfite Conversion and Pyrosequencing

To determine the degree of DNA methylation of the *S100a8* gene, pyrosequencing was performed employing specifically designed primers flanking the promoter region of *S100a8*. Briefly, 0.5 µg of genomic DNA was bisulfite-modified using the EZ DNA Methylation Kit (Zymo Research, Irvine, CA, USA), and PCR amplification of target regions was performed with the PyroMark PCR Kit (Qiagen, Venlo, The Netherlands). The biotin-labeled PCR products were captured by Streptavidin–Sepharose HP (Amersham Pharmacia. Amersham, UK), then purified and made single-stranded using a Pyrosequencing Vacuum Prep Tool. The sequencing primers were then annealed to the single-stranded PCR product. PyroMark Q24 (Qiagen, Venlo, The Netherlands) was applied for subsequent pyrosequencing reactions and methylation quantification. The forward and reverse primers for the amplification of the *S100a8* promoter were as follows: cg1-PCR_F, TTTGGATTGGGAAATTGGTTGA; cg1-PCR_R, Biotin-ACTTATTCCCAAAAATTCATAAATACAATC; cg2-PCR_F, AGTTGAGGGTGGATAGAATGAA; cg2-PCR_R, Biotin-TCCCAAAAATTAAACAAATTCCTAACCTA; cg3-PCR_F, AGGAAATAGAGGTTGTGGTAATT; cg3-PCR_R, Biotin-CTAAACACTTCCTACCCTCTTC. The sequencing primers used were as follows: cg1-PCR_S1, TTTTTTATTTTTTTGGGTTTTAGT; cg1-PCR_S2, AGATTTTTATAAATTGGTAGT; cg2-PCR_S1, TTTTAGTTTTTAGAGTAGAATGATT; cg3-PCR_S1, GGTTGTGGTAATTTTGG; cg3-PCR_S2, AGTTTTATATATTTTTTGTTAGTT; cg3-PCR_S3, GGGATAGAGTAGTTTTTTTTG.

### 4.7. Study Population and Mutational Analysis

Patients with Ph(-) MPN who were followed at our institute were enrolled in this study. Adult healthy individuals were also included as controls. The study was approved by the Institutional Review Board of Chang-Gung Memorial Hospital (IRB approval number: 201901412B0C502). All participants provided informed written consent in accordance with the Declaration of Helsinki. The detection of *JAK2*V617F and *CALR* exon 9 mutations in clinical samples was performed as previously described [38].

### 4.8. Statistics

The two-tailed independent Student’s *t*-test was used to compare continuous variables between two groups. For comparison between the dichotomous variables, a Pearson chi-square or a Fisher’s exact test (for expected values of >5 or ≤5, respectively) was applied. The Kaplan–Meier method and Log–Rank test were adopted to compare survival between patient groups. The Statistical Package of Social Sciences 18.0 (SPSS, Inc., Chicago, IL, USA) and GraphPad Prism 7.0 (GraphPad Software, Inc., San Diego, CA, USA) were employed to perform all the statistical calculations. The numerical results were presented as the mean ± SD. The horizontal lines in the dot plots represented the mean value of each group. Asterisks in graphs symbolize the significance of *p* values comparing the indicated group with controls unless specifically indicated (*, *p* < 0.05; **, *p* < 0.01; ***, *p* < 0.001, ns, no significance).

## Figures and Tables

**Figure 1 ijms-24-08747-f001:**
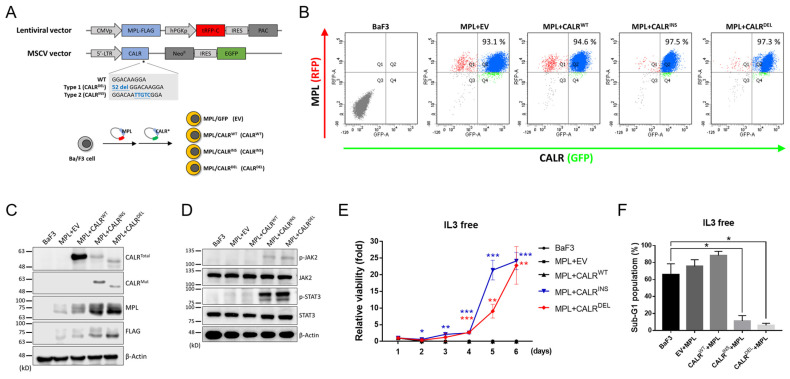
**Examination of established stable MPN clones containing *CALR*-exon 9-indel mutations.** (**A**) A schema depicting the viral vectors (upper panel) and the transduction procedures (lower panel). A human *MPL* sequence tagged with FLAG was cloned into an RFP-containing lentiviral vector, whereas human *CALR* exon 9 sequences [either wild-type (WT), type 1 mutant (52bp deletion), or type 2 mutant (5bp insertion)] were cloned into an MSCV vector harboring GFP fluorescence. Ba/F3 cells were transduced with both vectors to obtain four different clones. EV: empty vector. (**B**) Flow cytometric analysis demonstrating the obtainment of desirable dual RFP^+^GFP^+^-expressing cells. (**C**) Western blots on the expression of MPL, FLAG, and different forms of CALR proteins in various stable cell lines. (**D**) Western blots on the expression levels of proteins involved in the JAK2/STAT3 signaling pathway in the indicated cell lines. (**E**) Time-dependent proliferation of various stable cells in the absence of IL-3, as measured by manual counting. (**F**) Quantification of sub-G1 populations in each cell line. The content of DNA was analyzed with flow cytometry after Propidium Iodide (PI) staining. *: *p* < 0.05; **: *p* < 0.01; ***: *p* < 0.001 by Student’s *t*-test.

**Figure 2 ijms-24-08747-f002:**
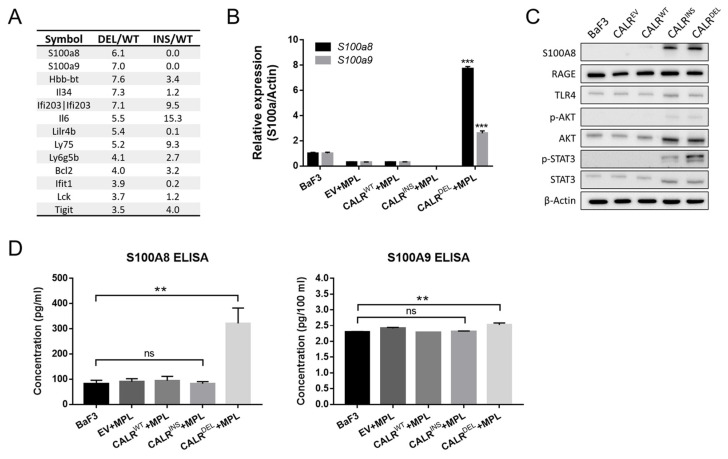
**The expression of *S100a8/a9* in *CALR*-mutated MPN-model cells.** (**A**) Comparison of the expression levels of genes between *CALR*^DEL^ and *CALR*^WT^ (DEL/WT) cells and between *CALR*^INS^ and *CALR*^WT^ (INS/WT) cells through RNA-Seq analysis. Only representative genes with differential expression between *CALR*^DEL^ and *CALR*^WT^ (DEL/WT) cells were shown. (**B**) Quantitative RT-PCR analysis of *S100a8/a9* transcript levels in parental BaF3 cells, stable BaF3 cells co-transfected with *MPL* and either empty vector (EV), *CALR*^WT^, *CALR*^INS^, or *CALR*^DEL^. (**C**) Western blots on the expression levels of S100A8 and the downstream effectors in the indicated cell lines. (**D**) Quantification of S100A8 and S100A9 levels in the cell culture supernatants of various cells, as assessed with ELISA. **: *p* < 0.01; ***: *p* < 0.001; ns: no significance by Student’s *t*-test.

**Figure 3 ijms-24-08747-f003:**
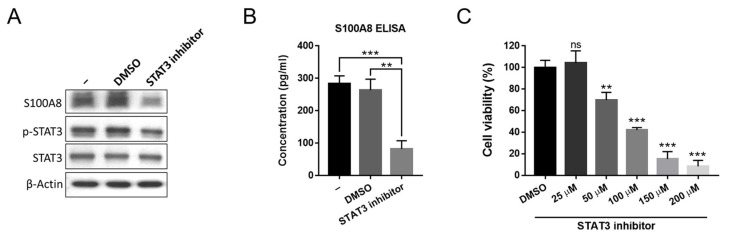
**Effects of STAT3 inhibition on S100A8 expression in *CALR*^DEL^ cells.** (**A**) The expression levels of S100A8, phosphorylated STAT3 (p-STAT3), STAT3, and Actin in *CALR*^DEL^ cells were subjected to various treatments. Untreated cells and cells treated with either DMSO or STAT3 inhibitor S3I-201 at a concentration of 25 μM for 48 h were harvested and subjected to western blot analysis. (**B**) Quantification of secreted S100A8 in *CALR*^DEL^ cells. Supernatant from the aforementioned cell culture was used for ELISA analysis. (**C**) Assessment of the impacts of various concentrations of the STAT3 inhibitor on viability of *CALR*^DEL^ cells. Cells were treated with either DMSO or increasing levels of S3I-201 in an XTT assay. **: *p* < 0.01; ***: *p* < 0.001; ns: no significance by Student’s *t*-test.

**Figure 4 ijms-24-08747-f004:**
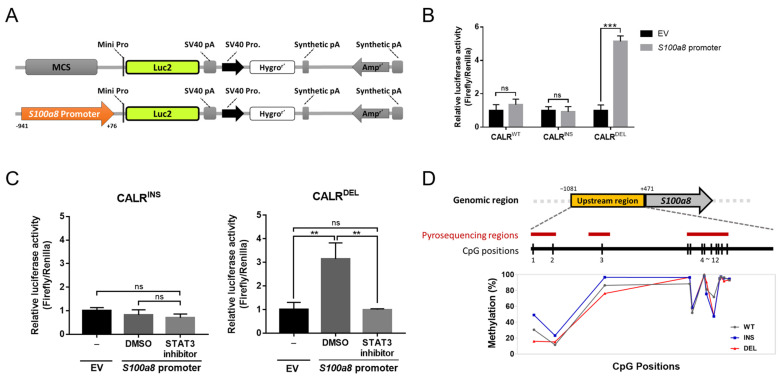
**Identification of *S100a8* as a STAT3 target in *CALR*^DEL^ cells.** (**A**) Schematic diagram of viral constructs for luciferase reporter assay. The luciferase reporter empty vector (EV) was used as control (upper panel), and the *S100a8* promoter sequence was cloned into the 5′ end of the luciferase gene in this vector (lower panel). (**B**) The luciferase activity was induced by either the EV control or the vector cloned with *S100a8* promoter sequence in *CALR*^WT^, *CALR*^INS^, and *CALR*^DEL^ cells, respectively. (**C**) The luciferase activity in *CALR*^INS^ cells (left panel) and *CALR*^DEL^ cells (right panel), respectively. Cells were treated with either EV or the vector cloned with *S100a8* promoter sequence, with the latter one further exposed to either DMSO or the STAT3 inhibitor S3I-201 at a concentration of 25 μM for 48 h before being subjected to analysis. (**D**) Bisulfite pyrosequencing within the *S100a8* promoter region of three different cell lines. Upper panel: schematic diagram depicting the genomic map of the *S100a8* promoter (yellow color). The enlarged region illustrates the locations of CpG sites (black vertical lines) and regions for bisulfite pyrosequencing analysis (red lines). Lower panel: scatter plot demonstrating the degree of methylation within twelve CpG sites in *CALR*^WT^, *CALR*^INS^, and *CALR*^DEL^ cells. **: *p* < 0.01; ***: *p* < 0.001; ns: no significance by Student’s *t*-test.

**Figure 5 ijms-24-08747-f005:**
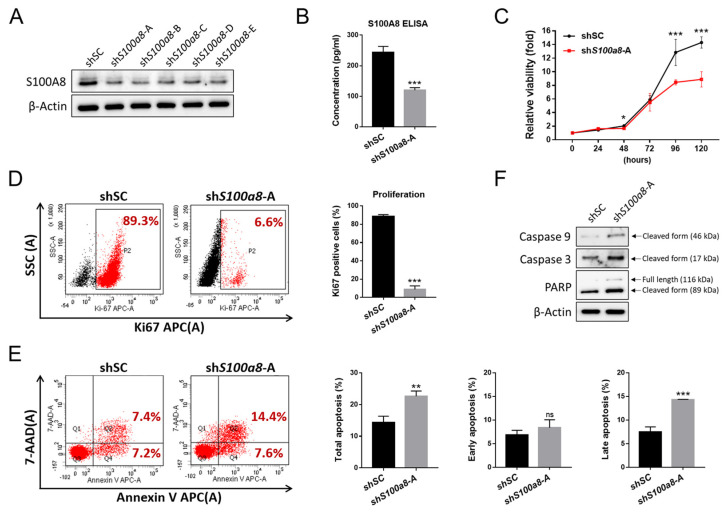
**Involvement of S100A8 in cellular proliferation and survival in *CALR*^DEL^ cells.** (**A**) Western blots on the expression levels of S100A8 in *CALR*^DEL^ cells treated with shRNAs specific to *S100a8* (sh*S100a8*; clones (**A**–**E**)) or a scramble sequence (shSC). Actin was used as a loading control. (**B**) ELISA analyses on the levels of secreted S100A8 in *CALR*^DEL^ cells exposed to either shSC or sh*S100a8*-A. (**C**) The viability of *CALR*^DEL^ cells receiving shRNAs, as measured with XTT assay. (**D**) Flow cytometric analyses on cellular proliferation index of shRNA-treated *CALR*^DEL^ cells, assessed with Ki67 staining. (**E**) Rate of apoptosis in *S100a8*-knockdown *CALR*^DEL^ cells, in comparison with control cells. *CALR*^DEL^ cells receiving shRNAs were treated with 7-AAD and APC-Annexin V and subjected to flow cytometric analyses. Early- and late-stage apoptosis together accounted for total apoptosis. Cells positively stained for Annexin V only were defined as being in early apoptosis, whereas those with positive stains for both Annexin V and 7-AAD were considered as being late apoptotic cells. (**F**) Western blots on the expression levels of various apoptosis-related proteins, including cleaved caspase 9, cleaved caspase 3, full-length and cleavage form of poly (ADP-ribose) polymerase (PARP) in *CALR*^DEL^ cells treated with either shSC or sh*S100a8*-A. *: *p* < 0.05; **: *p* < 0.01; ***: *p* < 0.001; ns: no significance by Student’s *t*-test.

**Figure 6 ijms-24-08747-f006:**
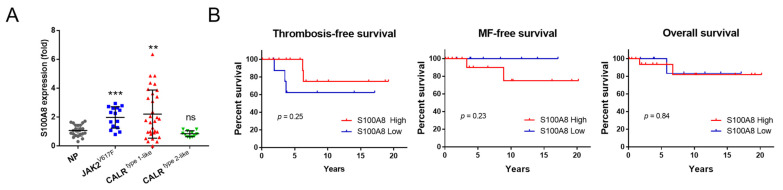
**Validation of significance of *S100A8* expression in clinical MPN patients.** (**A**) Quantification of *S100A8* expression in clinical samples. *S100A8* expression in the peripheral blood granulocytes of MPN patients and normal population (NP) was assessed with qRT-PCR. MPN patients were stratified according to their driver mutation status. **: *p* < 0.01; ***: *p* < 0.001; ns: no significance by Student’s *t*-test. (**B**) Kaplan–Meier estimates of survival outcomes in type 1-like *CALR*-mutated MPN patients stratified by the expression level of the *S100A8* gene. Log–Rank test was employed to assess its prognostic significance on thrombosis-free survival, MF-free survival, and overall survival.

**Table 1 ijms-24-08747-t001:** Clinical and laboratory features of type 1-like *CALR*-mutated MPN patients, stratified by the expressional level of *S100A8* gene.

	*S100A8*^High^ N = 21	*S100A8*^Low^ N = 10	*p* Value
**Age, years**	62.8 ± 16.5	56.6 ± 22.2	0.383
**Male, no. (%)**	12 (57.1%)	7 (70.0%)	0.492
**Subtype**			0.242
**ET**	16	10	
**PMF/PrePMF**	5	0	
**Splenomegaly**			0.673
**Yes**	5	4	
**No**	8	4	
**WBC count, ×10^9^/L**	10.6 ± 6.5	9.3 ± 3.1	0.458
**Hemoglobin, g/L**	11.5 ± 2.3	12.5 ± 1.7	0.216
**Hematocrit, %**	35.0 ± 6.7	38.4 ± 4.7	0.163
**Platelet, ×10^9^/L**	625 ± 318	975 ± 527	0.077
**LDH, U/L**	325 ± 130	257 ± 70	0.285
**Uric acid, mg/dL**	6.3 ± 1.5	7.0 ± 1.4	0.389
**Major thrombosis, no. (%)**	5 (23.8%)	3 (30.0%)	0.713
**Post-ET MF transformation, no. (%)**	2 (12.5%)	0 (0%)	0.260
**AML transformation, no. (%)**	1 (4.8%)	0 (0%)	0.483

Abbreviation: ET: essential thrombocythemia; PMF: primary myelofibrosis; WBC: white blood cell; LDH: lactate dehydrogenase; AML: acute myeloid leukemia. Values are reported as mean ± SD or number (with percentage in parentheses).

## Data Availability

All relevant data are available from the corresponding authors upon reasonable request.

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
