# Peer review of "Mutation-Driven S100A8 Overexpression Confers Aberrant Phenotypes in Type 1 CALR-Mutated MPN"

_ijms, 2023, doi:10.3390/ijms24108747_

Round 1
Reviewer 1 Report (Previous Reviewer 1)
The authors have addressed most of the comments of the first review. Still, there some minor comments/clarifications to be addressed:
line 206: please replace the word "transduction" with the word "transfection" (if virus was not used as the authors describe in M&M..). That is why my comment on the first review!
line 258/259: please remove the word "profound" (Figure 5E: 22% of the shRNAS100A8 versus 14,6% on scrambled is not profound apoptosis, it is just 7,o% higher of dying cells). Please describe when your flow cytometry analysis was performed after transduction with the lenti-virus(its not described in the M&M neither in the results).
line 263: please remove the word "indispensable". Cells are not affected dramatically from the S100A8 reduced expression by shRNAS100A8. Please tone down your results.
line 365-366; Replace "between the two cells" with "between the two types of mutated cells" or similar...
line 370: replace the " possible reinforce" with " they are in concordance with"or similar...
Minor editing(see comments).
Author Response
line 206: please replace the word "transduction" with the word "transfection" (if virus was not used as the authors describe in M&M..). That is why my comment on the first review!
Response: Thank you for your valuable reminders, and we sincerely apologize for any confusion that may have arisen from our previous wording. We have revised the text to address your concerns.
line 258/259: please remove the word "profound" (Figure 5E: 22% of the shRNAS100A8 versus 14,6% on scrambled is not profound apoptosis, it is just 7,o% higher of dying cells). Please describe when your flow cytometry analysis was performed after transduction with the lenti-virus (its not described in the M&M neither in the results).
Response: Thanks for the kind suggestion. We appreciate your feedback and agree that the use of the word "profound" might have been too extreme. Therefore, we have revised the wording accordingly. Regarding the transduction of shRNA with the lentivirus, we realized that we did include a description in the last paragraph of M&M 4.1. Additionally, we have updated the figure legend in Fig. 5E to make it clear that the flow cytometry analysis was performed after transduction with the lentivirus. Thank you for bringing this to our attention.
line 263: please remove the word "indispensable". Cells are not affected dramatically from the S100A8 reduced expression by shRNAS100A8. Please tone down your results.
Response: Thanks for the kind suggestion. We have revised our manuscript by replacing the word "indispensable" with "important" to more accurately reflect the significance of our findings. Thank you for your valuable feedback.
line 365-366; Replace "between the two cells" with "between the two types of mutated cells" or similar...
Response: Thanks for the kind suggestion. We revised the wording accordingly.
line 370: replace the " possible reinforce" with " they are in concordance with"or similar...
Response: Thanks for the kind suggestion. We revised the wording accordingly.
Reviewer 2 Report (Previous Reviewer 2)
No further comments
Author Response
Thank you for taking the time to review our manuscript. We do appreciate your careful consideration and valuable support. The encouragement is truly motivating, and we look forward to continuing our research on related topics.
This manuscript is a resubmission of an earlier submission. The following is a list of the peer review reports and author responses from that submission.
Round 1
Reviewer 1 Report
Ying-Hsuan Wang et al. describe the effect of S100A8/9 differential expression in CALR mutants (Type I &II) O/E BAF3 cells. They show that the increased expression of S100A8 in CALR Type I O/E BAF3 cells is specific and not found in the CALR Type II O/E BAF3 cells after performing RNAseq on established cell lines. They also show that there is a respective increase in the S100A8 secreted protein in the supernatant of the CALR Type I O/E BAF3 cells only even though both CALRmutant O/E BAF3 cell lines present JAK/STAT activation (Western blot analysis). Further, they show that inhibition of the JAK/STAT activation by STAT3 inhibition reduces the protein levels of S100A8 in CALR Type 1 O/E cells and consequently the secreted protein. They also claim that this is mediated by STAT3 on a luciferase assay and they have performed bisulfite pyrosequencing to show that S100A8 promoter is hypomethylated in CALR Type I O/E BAF3 cells. Additionally, by knockdown of the S100A8 protein they also show that there is increased apoptosis in CALR Type I O/E BAF3 cells.
The authors have been following previous work on the S100A8/9 trying to establish a link in the differential expression of the S100A8 and S100A9 proteins and the clinical discrepancies among MPN patients carrying different mutations, specifically between CALR Type I and Type II MPN patients.
As it is demonstrated already from Fig.1E the viability of the CALRmut O/E cells in IL3 free conditions is low, although higher than the rest of te established cell lines. My concern in this type of experiments is that the mutant CALR is unstable and rapidly degraded as has been demonstrated previously. Therefore the results of all subsequent experiments (Fig 2, 3, 4,5) are dependent on the actual expression of the CALR mutants, which actually is not shown (please comment on that).
In Fig 2 RNAseq data are shown regarding the up regulation of S100A8/9 in CALR Type I O/E BAF3 cells but the protein levels are not clear for the S100A9 (Fig 2C).
In Fig 3. it is selectively presented the inhibition of STAT3 in CALR Type I O/E cells. Why? Since they both present JAK/STAT activation (Fig1D). Please show the data on CALR Type II O/E BAF3 cells. Is it the same?
In Fig 4.C the major difference in the luciferase activity is found due to the increase (3 fold) in the DMSO sample in the CALR DEL BAF3 cells as compared to the CALR INS BAF3 cells. Please comment....
In Fig5. the knockdown of S100A8 causes increased apoptosis in the CALR DEL O/E BAF3 cells. This experiment has been performed in IL3 free conditions? What are the actual numbers of cells that are viable there? Still, as mentioned before the CALR DEL protein levels are not shown....
In section 2.6 the authors compare CALR mutant MPN patients (Type -I like vs Type II-like). Why they do not actually compare CALR Type I vs CALR Type II specifically that are the ones that cause the disease (driver mutations)? Also they mistakenly claim that there is a difference in PLT counts among the S100A8-high and S100A8-low group (p-val is not significant). This section should be definitely revised.
Finally, the discussion is rather chaotic and a bit confusing without actually linking the presented data in the context of the initial scope of their study. It should be somewhat reduced and rewritten focusing on the impact of their results and not making a general description of what is known in the literature about S100A8/9.
In general, the work is important but the missing link to the physiology of the disease and the actual role of the S100A8 is not being forward in the present status of the manuscript.
Reviewer 2 Report
The authors examined the occurrence and effects of S100A8 overexpression in the pathogenesis of myeloproliferative neoplasmas using mainly a manipulated cell line as model system. The data are original and novel and appear to be of interest of any reader of the article.
Specific Points of Criticism and Suggestions for Alterations:
(1) Line 152, line 176, line 221, line 343 and elsewhere: "MPN cells" is wrong, it should read "in BaF3 cells" (or similar). The authors equate wrongly the manipulated BaF3 with „MPN cells“; the murine BaF3 cells are clearly not real human MPN cells and can only be used as a substitute, namely a model system.
(2) In general the English of the manuscript is acceptable, however it would nevertheless benefit from some English editing.
Suffice to mention here a few examples (among several others):
-- line 202, line 242, line 427: „endued“ is not appropriate, suggestion „induced“
-- line 209: „bewildering, suggestions „interesting“
Round 2
Reviewer 1 Report
The authors have responded to the comments of the 1st review by explaining better their methodology and recognising the cavities of their experimental approaches. However, their explanations are not totally satisfactory and they still fall short as they do not provide any new experimental evidence. Specifically on the comment about the degradation of the mutant CALR they recognise that is absolutely crucial but clearly they have not paid attention on their experimental design throughout their study. The western blot analysis provided is not clear whether it's a new or an old experiment and at which exactly stage or assay. But most importantly they keep on defending their transfection efficiency as a guarantee of the presence of the mutant protein, which is a very weak argument.
Regarding Point 2, they claim that the S100A9 protein detection is difficult but then they keep on interpreting the data of this blot and in particular of this protein in their text (lines 149-152). If the western is not conclusive then it has to be removed and the comments have to be adjusted regarding this particular protein.
Point 3. They refer to the low protein levels of S100A9 (western blot) as a reason to focus only on S100A8 in CALR DEL cells while previously they presented an inconclusive western blot. They do not provide any data on JAK-STAT pathway inhibition on CALR INS cells...at least to show secreted levels of S100A9, which can be detectable by ELISA. The fact that the protein is hardly detectable by western does not mean that small differences at the protein level upon JAK-STAT inhibition do not have any downstream effect. Please comment on your text...
Point 4. It is still unclear why the DMSO treatment causes a 3-fold increase in the luciferase activity as compared to EV in CALR DEL cells, while no difference in CALR INS cells....please comment on that. These are co-transfected experiments with the CALR mutants (not showing expression as mentioned before...) and the luciferase virus vectors. What if the transduction with the virus has impaired the expression of CALR mutants? Please comment on that....
Point 5. The downregulation of shRNA for S100A9 is not shown even though the expression at the RNA levels is similar to the S100A8 (fig 2A) in CALR DEL cells. Please comment on that new observation if possible...
Point 6. The explanation of the authors is not sufficient at all. The parameters on table 1 are not significant at all according to the p-val and this should be reflected accordingly in the text, although better be removed completely because they have no relevance and the section on clinical significance of S100A8 low vs high is totally speculative.
Overall, the authors have not adequately addressed the major points of the 1st review comments and they should try to improve their manuscript.